# An Investigation about Modern Deep Learning Strategies for Colon Carcinoma Grading

**DOI:** 10.3390/s23094556

**Published:** 2023-05-08

**Authors:** Pierluigi Carcagnì, Marco Leo, Luca Signore, Cosimo Distante

**Affiliations:** 1Institute of Applied Sciences and Intelligent Systems (ISASI), National Research Council (CNR), Via Monteroni snc University Campus, 73100 Lecce, Italymarco.leo@cnr.it (M.L.); 2Dipartimento di Ingegneria per L’Innovazione, Università del Salento, Via Monteorni snc University Campus, 73100 Lecce, Italy; luca.signore@studenti.unisalento.it

**Keywords:** colon carcinoma, artificial intelligence, deep learning, ensembling, histopathology

## Abstract

Developing computer-aided approaches for cancer diagnosis and grading is currently receiving an increasing demand: this could take over intra- and inter-observer inconsistency, speed up the screening process, increase early diagnosis, and improve the accuracy and consistency of the treatment-planning processes.The third most common cancer worldwide and the second most common in women is colorectal cancer (CRC). Grading CRC is a key task in planning appropriate treatments and estimating the response to them. Unfortunately, it has not yet been fully demonstrated how the most advanced models and methodologies of machine learning can impact this crucial task.This paper systematically investigates the use of advanced deep models (convolutional neural networks and transformer architectures) to improve colon carcinoma detection and grading from histological images. To the best of our knowledge, this is the first attempt at using transformer architectures and ensemble strategies for exploiting deep learning paradigms for automatic colon cancer diagnosis. Results on the largest publicly available dataset demonstrated a substantial improvement with respect to the leading state-of-the-art methods. In particular, by exploiting a transformer architecture, it was possible to observe a 3% increase in accuracy in the detection task (two-class problem) and up to a 4% improvement in the grading task (three-class problem) by also integrating an ensemble strategy.

## 1. Introduction

Cancer is a condition where cells in a specific part of the body grow and reproduce uncontrollably. Finding cancer (the detection task) early, when it is small and has not spread, often allows for more treatment options. After a cancer diagnosis, staging (the grading task) provides essential information about the extent of the cancer in the body and the anticipated response to treatment [1]. The observation of the cellular phenotype is a crucial point in cancer diagnosis. Grading describes how cancer cells look compared to normal, healthy cells. Knowing the grade gives the healthcare team an idea of how quickly the cancer may be growing and how likely it is to spread. Detection and grading were traditionally carried out by expert pathologists by directly observing tissue samples under a microscope. Nowadays, glass slides are converted into digital slides that can be viewed, managed, shared, and analyzed on a computer monitor. Nevertheless, interpreting histological images is time-consuming and prone to errors [2]: the primary sources of bias are the image quality, the expertise of the observer, and the multiple tissue types and characteristics. Developing computer-aided approaches for cancer diagnosis and grading is currently receiving an increasing demand: this could take over intra- and inter-observer inconsistency, speed up the screening process, increase early diagnosis, and improve the accuracy and consistency of the treatment-planning processes [3,4,5]. Cancer sometimes begins in one part of the body before spreading to other areas. Cancers can be grouped according to the type of cell they start from. There are several types of cancer and there are differences in detection and staging methods due to the way in which different cancers arise and spread. The most common type is carcinoma. Carcinoma is a type of cancer that begins in the epithelial tissue, a type of body tissue that forms the covering of all internal and external surfaces of your body, lines body cavities and hollow organs, and is the major tissue in glands. In this paper, the focus is on colorectal carcinoma (CRC), a disease in which cells in the colon or rectum grow out of control. As the third most common malignancy and the second most deadly cancer, CRC induced an estimate of 1.9 million incidence cases and 0.9 million deaths worldwide in 2020 [6]. CRC is usually divided into three grades: well-differentiated (low-grade), moderately differentiated (intermediate-grade), and poorly differentiated (high-grade) [7]. Carcinomas representing higher grades or poor differentiation tend to grow and spread faster. In the last years, several works have introduced modern machine learning strategies for making computer-aided approaches for CRC diagnosis and grading more reliable. Most of them rely on deep learning (DL), and convolutional neural networks (CNN). It is well known that each CNN model has a different way to represent data and to extract knowledge for generalization and prediction. It is quite common to observe correct predictions from a model that, on the other hand, fails on data on which others are robustly correct, and vice versa. In addition, CNN models often fail to learn long-range dependencies that are important in medical images, since very large images refer to the same tissue and then distant pixels are correlated. This is due to the inherent functioning, since they work at a pixel level, identifying features such as corners or lines by building their way up from the local to the global. To overcome these problems, researchers in machine learning have introduced the ensemble process and, more recently, transformer architectures. Ensemble [8] methods combine prediction outcomes from different models to reduce the variance, thereby improving the accuracy, of an automated decision-making system. They have been successfully used to address a variety of machine learning problems, such as feature selection, confidence estimation, missing features, incremental learning, error correction, class-imbalanced data, and learning concept drift from non-stationary distributions, among others [9]. In the field of medical image analysis, several ensemble strategies have been already successfully exploited on different benchmarks for COVID-19 and pneumonia classification, skin lesion classification, detection of rethynopatia, and tissue classification [10,11]. Transformer architectures exploit self-attention mechanisms to make connections between distant image locations [12]. Transformers offer multiple advantages, including design simplicity, robustness, and state-of-the-art performance on many learning tasks. Transformer-based network architectures have been introduced [13] for language tasks and, subsequently, they have been effectively adopted for improving image segmentation [14]. In the field of medical image analysis, transformers have also been successfully used in full-stack clinical applications, including image synthesis/reconstruction, registration, segmentation, detection, and diagnosis [15]. Unfortunately, from the study of the literature, it emerges that there are no works exploiting ensemble processes and integrating transformers into algorithmic pipelines aimed at colon cancer detection and grading. This paper fills this gap by testing a transformer architecture and subsequently several ensemble strategies on the largest publicly available dataset containing annotated histopathological images of the colorectal region. The experiments demonstrated that this allows professionals to shorten the training process and improves the ability of the models to automatically detect and grade CRC. Comparisons with leading state-of-the-art approaches showed a substantial improvement (up to 4%) in accuracy, paving the way in this challenging area for actual clinical applications of machine learning-based approaches. The rest of the paper is organized as follows: Section 2 describes related work whereas Section 3 and Section 3.4 introduce the methods and data, respectively. Then, Section 4 reports experimental results about grading colon carcinoma using transformers and convolutional networks. Finally, Section 6 concludes the paper.

## 2. Related Work

Automatic CRC detection and grading in histopathological images have not been deeply investigated yet. This is mainly due to the scarcity of annotated datasets. Building good datasets is a hard and time-consuming task: even highly skilled clinicians may disagree with one another on diagnosis and even more on defining grading levels.

Early works relied on handcrafted features and shallow classifiers [16,17,18]. The advent of deep learning has been a strong accelerator of research for automatic diagnosis of colon carcinoma [19] indeed. CNN has been recently exploited for this aim by providing small image patches as input (e.g., 224 × 224) and then aggregating predictions to obtain a diagnosis at the histological image level [20]. An intermediate tissue classification has sometimes been proven effective to help in improving accuracy [21]. In addition, different convolutional and recurrent architectures have been combined [22]. Since defining an appropriate image patch size is not trivial, some authors have tried to overcome this drawback by embedding some contextual information in the model [23] by sharing features across scales and then learning dependencies between scales using the long-short-term-memory (LSTM) unit. Differently, a cell-graph convolutional neural network (CGC-Net) has been exploited in [24] to model nuclear features along with their cellular interactions in the form of a graph, which accounts for both cell-level information and the overall tissue micro-architecture. More recently, an approach based on two stacked convolutional networks has been also introduced and tested. The first network is used for learning the histology image’s local representation, which is then aggregated considering its spatial pattern by the second network [25]. A recent study [26] compared several CNN architectures and has demonstrated that classical network models, designed for the task of image classification, work better than introducing domain-specific solutions due to the lack of data for a robust knowledge generalization. Finally, recent studies have demonstrated that an attention mechanism, added in parallel to capture key features that facilitate network classification, can really help CNN in accomplishing knowledge-extraction tasks from large histological images [27,28].

## 3. Method and Data

In this section, the different strategies used in this paper to improve CRC detection (two-class problem, i.e., healthy tissue/cancer) and grading (three-class problem: low, medium, or high carcinoma) are described. All the listed methodologies have never been exploited for colon carcinoma grading. In addition, the datasets used for training and testing the proposed pipelines are described.

### 3.1. Convolution Neural Networks (CNN)

#### 3.1.1. ResNet

The various architectural proposals introduced over the years have highlighted the use of an increasing number of convolutional layers, arranged in a cascade, in order to improve performance [29]. While increasing depth has resulted in the better extraction of high-level features, the problem of the vanishing gradient arises when too many layers are stacked [30,31]. In particular, if for networks with tens of layers the problem has been tackled by resorting to techniques based on the normalization of the intermediate layers [32,33], for deeper networks a degradation problem emerges, not attributable to overfitting, which leads to a greater training error [34]. To overcome this degradation problem, the ResNet architectures [35] introduce a deep residual learning approach in which shortcut identity connections, between convolutional layers, are exploited in order to strengthen the information flow. This has been shown to effectively mitigate the vanishing gradient problem, thus allowing the use of very deep networks with non-negligible gains in accuracy.

In this work, the ResNet50 network, where the number indicates the number of layers that can actually be trained, was tested. Its architecture is built up of four stages: the starting channel width is set to the size of input images and then the convolution and max-pooling use 7×7 and 3×3 kernel sizes, respectively.Afterwards, Stage 1 of the network starts, and it has three residual blocks containing three layers each. The size of kernels used to perform the convolution operation in all three layers of the block of Stage 1 are 64, 64, and 128, respectively. As we progress from one stage to another, the channel width is doubled and the size of the input is reduced to half.

#### 3.1.2. DenseNet

In order to further mitigate the vanishing gradient problem, in DenseNet [36] architectures, taking advantage of the skip-connection approach introduced with ResNet, each layer is connected to every other layer in a feed-forward fashion, leveraging the re-use of features.Unlike ResNet, where features maps are combined by summation before being passed as input to the layers, in the DenseNet architecture concatenation of feature maps is used. The proposed connection strategy requires fewer parameters than a corresponding traditional CNN and involves a strengthened feature propagation alleviating the vanishing-gradient problem for very deep implementations. In this work, the DenseNet121 architecture was tested. It has 1 initial convolution and pooling layer, 3 transition layers, 5 dense blocks (1×1 and 3×3 conv) layered 6,12,24,16, respectively, and a final fully connected classification layer. Summing up, the model has [1+(6+12+24+16)×2+3)+1=121] learnable layers, as indicated in the name. The hyperparameter *k*, called growth rate, helps to generalize the *l*th layer in the following manner: k[*l*] = (k[0] + k[*l*−1]), where k[0], known as the number of channels, was set to 32.

#### 3.1.3. SENet

A different approach has been taken with the proposed SENet architecture [37]. In particular, techniques aimed at better highlighting the correlations in the feature space were studied. This led to the introduction of a new module called *squeeze and excitation* (SE) that can explicitly model the interdependencies in the convolution layers so that the network, during the training phase, learns to emphasize the most information at less expense.SENet is mainly composed of three basic parts: (1) The squeeze operation is a compression step that uses global average pooling to pool the input tensor into specific data. (2) The excitation operation, which generates different weights to act on each channel and constructs the dependencies between different channels; a fully connected multi-layer perceptron (MLP) bottleneck structure is used to map the scaling weights. The MLP has a single hidden layer along with the input and output layers, which are of the same shape. The hidden layer is used as a reduction block where the input space is reduced to a smaller space defined by the reduction factor (which is set at 16). The compressed space is then expanded back to the original dimensionality as the input tensor. (3) The reweight operation (scaling), which receives the "excited" tensor from the excitation module, and through a sigmoid activation layer, scales the values to a range of 0–1. Subsequently, the output is applied directly to the following input layer by a simple broadcasted element-wise multiplication, which scales each channel/feature map in the input tensor with its corresponding learned weight from the MLP in the excitation module. The SE blocks can be integrated into existing network architectures, such as ResNet, or be arranged in a cascade, thus obtaining a different architecture (SENet). In this work, various implementations have been investigated and, for the problem faced, the best results have been obtained with the SE-ResNet50 network by integrating the SE blocks into the ResNet50 network architecture. In particular, a bottleneck SE-ResNet module was implemented with one 3×3 convolution surrounded by dimensionality reducing and expanding 1×1 convolution layers, and then combined with the SE block. This way, each SE block was placed inside the residual unit, placing it directly after the 3×3 convolutional layer. Since the 3×3 convolutional layer possesses fewer channels, the number of parameters introduced by the corresponding SE block is also reduced. Its structure can be then resumed as: 1×1—conv, 3×3—conv, 1×1 SE-block.

#### 3.1.4. EfficientNet

Following a different investigation, in order to further improve state-of-the-art network architecture performances, in [38] the authors proposed a scaling method that aims to uniformly scale depth, width, and the resolution of a given network using a single *compound coefficient*. Starting from a baseline network with mobile-size complexity (EfficientNet-B0), a family of eight networks with increasing grades of complexity was built up. The base EfficientNet-B0 network is based on the inverted bottleneck residual blocks of MobileNetV2 [39], in addition to squeeze-and-excitation blocks. The other architectures in the family are similar to the above model; the only difference between them is that the number of feature maps (channels) varies according to the compound parameter, which incrementally changes the number of sub-blocks and, as a consequence, the number of parameters. In other words, each architecture contains 7 blocks. These blocks further have a varying number of sub-blocks whose number is increased as we move from EfficientNetB0 to EfficientNetB7. The number of layers ranges from 237 in the EfficientNet-B0 model to 813 in the EfficientNet-B7 model.

#### 3.1.5. RegNet

Instead of designing the best-performing individual network, a possible viable alternative is the exploitation of tools derived from classical statistics suitable to study the behaviour of populations of models and discover general design principles. This is the key idea behind the recent work in [1], in which authors proposed a framework aimed at exploring a design space where populations of networks can be parameterized. This process should lead to optimally designed networks called RegNet models. They are not defined by fixed parameters such as depth and width, but rather by a quantized linear function controlled by the chosen parameters. The starting point is an unconstrained generic ResNet Architecture with all possible values of parameters, i.e., D = depth, W0 = initial width, WA = slope, WM = width parameter, B = bottleneck, and G = group. The term width indicates the number of channels in each layer. The bottleneck ratio is used to reduce the number of channels of the input feature map and group size is used to crry out the parallel group convolutions. The networks obtained in the design space were called RegNet (regular networks). Different models with different computational loads and epochs can be generated. In particular, models with (RegNetY) and without (RegNetX) a squeeze-and-excitation (SE) block can be generated. In addition, any specific generated model is indicated by the corresponding FLOPs regime. Hence, for instance, RegNetY-400MF means that a model with an SE block and a computational regime of 400 Mega-FLOPs was built. In particular, in this work, a RegNetY architecture was built with 6.4 and 16 gigaflops and named RegNetY6.4GF and RegNetY16GF, respectively. The parameters chosen for the RegNetY6.4GF architecture were: D = 25, W0 = 112, WA = 33.22, WM = 2.27, B = 1, and G = 72. For the RegNetY16GF the parameters chose were: D = 18, W0 = 200, WA = 106.23, WM = 2.48, B = 1, and G = 112.

### 3.2. Transformer Networks

Deep convolutional neural networks have inherent inductive biases and they lack an understanding of long-range dependencies for understanding image content and learning representations that are highly expressive. Transformer-based architectures have been introduced [13] for language tasks and, subsequently, they have been effectively adopted for improving image segmentation [14]. In this paper, a transformer-based model endowed with an additional control mechanism in the self-attention module is exploited to understand discriminative regions in large histological images. The key idea is that for colon carcinoma grading, the transformer architecture can be of preliminary help to understand which areas of the images could help the following CNN architectures to discriminate among carcinoma grades. This way, models’ training could be performed using less data and in a more effective way. Figure 1 reports in a schematic way the pipeline, including the transformer architecture. Visual fields are given as input to a transformer network that combines local and global training in order to extract information from both the entire image and local patches in which finer details can be discovered [40]. The implemented transformer architecture is reported in Figure 2. The local–global training relies on shallow global and deep local branches. The whole visual field is processed by a conv block with three convolution layers (each followed by batch normalization and ReLU activation) which extracts a global feature map that is the input of the global branch. On the other hand, patches having one-quarter the size of the initial visual field are extracted and processed by a conv block identical to the one used for the whole visual field. This way, local feature maps are extracted. The feature maps are then provided as input to global and local branches, respectively. The global branch has two blocks of encoder and decoder, where the local branch instead has five blocks of encoder and decoder. In the local branches, the output feature maps are re-sampled based on their location to obtain the output feature maps. Each encoder block has a 1 × 1 conv layer followed by normalization and two layers of multi-head attention layers operating along the height and width axis, respectively. This implements the axial attention mechanism that avoids introducing independence assumptions and harvests the contextual information of all the pixels [41]. The attention map is then obtained by adding residual input maps to the concatenation of the output from the multi-head attention blocks after passing it through another 1 × 1 conv followed by normalization. Each decoder block consists of a conv layer followed by an up-sampling layer and ReLU activation. The transformer is trained in order to detect epithelium tissue, from which colon carcinoma starts, and which hence is considered an important bio-marker for tumour detection and grading [42]. After training, it is able to provide (also on unseen visual fields) the corresponding binary masks that point out glandular regions which can be then the only part retained for subsequent processing performed by CNN models.

### 3.3. Ensemble Approach

In the case of complex classification problems, as the considered one is, it is exceedingly difficult to build a learning model able to simultaneously address all the challenging issues. Sometimes it could be highly effective to train two or more models on the same data, and then combine their classification outcomes to obtain a more accurate final decision. This type of operation is reported in the literature as model ensembling [43], and the combination can be achieved by several strategies of combination such as majority voting, averaging, or other output combination approaches.

In this work, two ensemble strategies have been introduced for colon carcinoma grading. Each of them has been implemented in different ways by exploiting the best-performing architectures among those explained in previous subsections.

The first strategy, namely *max-voting ensemble*, combines for each patch the predictions of involved networks and it assigns to the patch the most voted label among all. As in the case of a single network model, a counting operation is subsequently performed to assign to the whole visual field the grade of the most numerous class among all the patches.

In Figure 3, a schematic representation of the *max-voting ensemble* strategy is depicted.

The second strategy, namely *Argmax ensemble*, works by computing an *Argmax* operation on the total number of patches for each class coming out from the combined networks. In other words, a vector of labels (having a length equal to the number of networks involved in the ensemble) is assigned to each patch. A counting operation is subsequently carried out on these vectors of labels and the class with the highest number of occurrences is assigned to the visual field.

In Figure 4, a schematic representation of the *Argmax ensemble* strategy is depicted.

### 3.4. Datasets

Two publicly available datasets of annotated histological images of colorectal tissues, namely Extended-CRC and GLAs, have been used to train and test the investigated pipelines, Extended-CRC is an extension of the well-known CRC-Dataset [18] which is comprised of 139 visual fields extracted from 38 hematoxylin- and eosin-stained whole-slide images (often abbreviated as H&E stained WSIs) of colorectal cancer cases. The visual fields have an average size of 4548×7520 pixels obtained at 20× magnification. Expert pathologists associated a cancer grade (normal, low, or high) with each visual field. The Extended-CRC dataset [44] complements the former dataset, with 161 additional larger visual fields extracted from 68 hematoxylin- and eosin-stained whole-slide images using the same criteria. The extended colorectal cancer (Extended-CRC) dataset then consists of 300 visual fields with an average size of 5000×7300 pixels. Table 1 reports a detailed distribution of the visual fields of different grades for both datasets. In this paper, the annotated visual fields in the Extended-CRC dataset have been used to train and test the different pipelines introduced in Section 3 and the related results are reported in Section 4.

The GLAs dataset [45] consists of 52 visual fields derived from 16 hematoxylin-and eosin-stained histological sections of colorectal carcinoma of stage T3 or T42. Visual fields were collected from 16 different users, and they show both benign and malignant pathologies. Each visual field was labelled by an expert pathologist as benign or malignant, and each glandular object on that visual field was manually segmented. In this paper, the provided boundaries of glandular regions have been used as the ground truth for training the transformer architecture.

## 4. Experimental Results

In the following subsections, the capability of methods introduced in Section 3 to automatically detect and grade visual fields in the Extended-CRC dataset are reported. According to the prevailing approaches in the literature, a three-fold cross-validation was carried out. The fold splitting provided with the Extended-CRC dataset was used. The choice of k was made to find a trade-off between the computational time for training the different models in such a complex task and the attempt to reduce the bias in estimating the true accuracy. To tackle the imbalance of classes in available data, a class weight approach, to allow the model to increase attention to samples from the minority class than the majority one, has been employed and exploited in the computation of the cross-entropy loss function during model training [46]. This way, classes’ contribution to the training process is balanced according to their data distribution. For cancer detection, the binary problem of distinguishing between normal and tumour tissues was addressed, whereas for cancer grading, the three-class problem of labeling the tissues as normal, low-grade, or high-grade was addressed instead. Non-overlapping patches of size 224×224 pixels were used, and a batch size of 16 was set for the training and classification steps. Figure 5 reports the cardinality and labeling content of each fold. Background class refers to patches without inner radiometric dynamics (in general with pixels all close to 255 value) which were not considered for processing.

The average accuracy was computed as the total amount of patches correctly classified on the total amount of processed patches. In a formula, for each fold *j* in range [1,K] (k=3 in the following experiments), *average accuracy* is computed as the average of:(1)$$acc_{j}=\frac{\sum_{i=1}^{c}{TP}_{i}}{\sum_{i=1}^{c}Ni}$$

Nonetheless, from Figure 5, it is straightforward to note that there is an unequal distribution of patches per class and computing only the above overall metric could be insufficient for an actual evaluation of the pipeline. For this reason, and according to the related literature, a second score was computed: namely, the weighted accuracy. The weighted accuracy was computed by considering the accuracy in every single class and then averaging the gathered scores. Using a formula, *weighted accuracy* is computed as the average of:(2)$$w\_acc_{j}=\frac{\sum_{i=1}^{C}\frac{{TP}_{i}}{N_{i}}}{C}.$$
where *C* indicated the number of classes (2 or 3), $N_{i}$ is the of elements in the class *i*, and $TP_{i}$ is the number of true positives for the class *i*.

All the investigated CNNs have been fine-tuned starting from the pre-trained models on the ImageNet dataset provided with the reference implementations.In particular, the stochastic gradient descent (SGD) optimizer was employed with learning rate = 0.001, momentum = 0.9, weight decay = 0.001 parameters, and a batch size of 16. A number of training epochs of 100 was set.

During the training phase of each architecture, the reduced learning rate on plateau scheduling was used. This approach adjusts the learning rate when a plateau in model performance is detected, e.g., no change for a given number of training epochs. Therefore, during the training, it monitors a quantity (loss) and if no improvements are seen for a patience number of epochs, the learning rate is reduced. In this experiment, the patience was set to 10.

Given the limited number of visual fields, some active data augmentation techniques were used: in particular horizontal and vertical flip, rotation by a random value sampled from the list [−90, −45, 45, 90], and shear on the x-axis by a random amount between −20 and 20 degrees.

For the transformer architecture, training configurations specified in [40] were used, i.e., a batch size of four, Adam optimizer, and a learning rate of 0.001. Finally, the network was trained for 400 epochs.

Experiments were performed on a workstation equipped with Intel(R) Xeon(R) CPU E5-1650 0 @ 3.20 GHz, GPU: GeForce GTX 1080 Ti, RAM-GPU: 11 GB, SO Ubuntu 16.04 Linux.

### 4.1. Transformer Model

In this section, the experimental results obtained by introducing a transformer network for segmenting glandular regions as a preliminary step are reported. The transformer was trained on images and binary masks provided by the GLAs dataset described in Section 3.4.

After training on the GLAs dataset, the learned configuration was exploited to a extract binary mask for the Extended-CRC dataset and then only patches corresponding to predicted glandular regions were used as input to the subsequent CNN-based colon carcinoma grading.

Figure 6 reports an example of how the transformer networks can help to provide an input to the subsequent classifiers-only patches related to glandular regions and the most useful ones for grading the colon carcinoma. In Figure 6a an original histological image containing a colon carcinoma of intermediate grade (grade 1) is shown. Figure 6b depicts the corresponding binary mask extracted by the transformer network and points out (white regions) glandular regions. The image obtained by an AND logical with the mash is then reported in Figure 6c and, finally, Figure 6d shows the patches retained for the subsequent steps in the CNN-based pipeline for carcinoma grading. It is worth noting how the transformer network puts attention on regions relevant for grading, making it possible to discard those patches that just introduce noise in the learning process.

After the aforementioned steps, the resulting patches’ distribution (extracted from the visual fields of the extended CRC dataset) per fold and class is reported in Figure 7. Of course, introducing this attention process discarded a significant number of patches, e.g., about 46% of the initial patches. Patches labelled as background increased (from 11% to more than 14% of the total number of extracted patches) since the radiometric values were checked after the application of the masks extracted by the transformer network. This helped to handle both sporadic noise regions and patches falling on the border of the remaining regions. This way, the subsequent CNN models worked on a percentage of 40% of the initial patches instead of 89% without preliminary transformer architecture.

It is worth noting that the number of visual fields to be classified (300 in the considered extended CRC dataset as reported in Table 1) did not change, but this just affected the number of patches that contributed to the final labeling of each of them. This way, only patches providing an actual contribution to the classification step were retained.

This claim was confirmed by the quantitative results in Table 2 that reports grading results by exploiting transformer networks for pointing out discriminative regions. According to [26], ResNet [35], DenseNet [36], SENet [37], EfficientNet [38], and RegNet [1] were exploited for classification.

The best classification results were obtained by combining the transformer and EfficientNet. In particular, the EfficientNet-B1 model was the best in binary classification whereas the EfficientNet-B2 model was the best in solving the three-class problem.

Table 3 reports the average training time with and without integrating transformer networks for pointing out discriminative regions and CNN architectures for detection and grading. As expected, the training times were much lower than those without the transformer net due to a reduced number of involved patches and their higher informative content. This is a highly critical aspect of this task, since it could allow using a larger amount of input examples, even exploiting augmentation strategies for facing the data scarcity problem. It also saves time and hardware resources when new data become available, and tuning is required to obtain more and more accurate models for enlarge trustworthy of methods. This also makes the application of explainability strategies easier, which is becoming essential in this research field for demonstrating how the model operates to the stakeholders.

It is useful to point out that the accuracy scores always refer to the whole set of visual fields in the Extended-CRC dataset. Therefore, gathered scores are comparable with all the experimental phases. The advantage of using a transformer is that it puts attention to significant patches for the final classification goals, discarding those with no features to help to build robust CNN models.

### 4.2. Ensembles

The best-performing networks in [26] (in terms of average and weighted accuracy on three classes) were exploited to build several ensembles. Considering recent studies on the effectiveness of the different ensemble strategies on medical imaging tasks [10], two simple statistical pooling functions were implemented. In Table 4, the lists of ensembles tested on the Extended-CRC dataset are shown. From left to right, the first column reports how the network combination has been labelled for subsequent references, the second column lists the involved network models, and the third column points out the mixing strategy used between the two described in Section 3.

From Table 5 it is possible to observe that the ensemble processes provided higher accuracies than those obtained in [26], where a single model was used for classification. The considered ensemble strategies perform similarly and it is possible to conclude that using one or the other is not a good choice. There is instead an appreciable difference in performance when the ensembled CNN models change: it is easy to see that the combination of EfficientNet-B7, RegNetY16GF, and SE-ResNet50 is the best solution among the investigated ones. It is also important to observe that, once again, introducing the transformer model to first select relevant regions is a key point for creating a big leap in accuracy. Ensemble strategies improve results by at least 2% with respect to the experimental tests without transformers.

## 5. Comparisons to Leading Approaches in the Literature

This section reports a comparison of the most performing pipelines introduced in this work with the leading approaches in the literature (data from previous works were taken from the original papers). Related average and weighted accuracy scores are similar (the same fold splitting provided together with the Extended-CRC was used in all the experiments) and are compared in Table 6, where it is possible to realize that the proposed solutions outperformed all previous ones, both for two-class (detection) and three-class (grading) classification tasks.

## 6. Conclusions

This paper proposes a transformer architecture and two ensemble strategies for automatically detecting (a classic binary classification problem) and grading (a more complex three-class problem) CRC through a machine learning approach. Experiments were carried out on the largest publicly available dataset containing annotated histopathological images of the colorectal region, and they demonstrated that the proposed solutions allow professionals to shorten the training process and improve accuracy for both clinical goals, i.e., the detection and grading of CRC. In particular, transformers, by exploiting embedded self-attention mechanisms, improved CNN classification results, since they focus on relevant areas. Ensembling several classifiers allowed us to lower the uncertainty of prediction.Comparisons with several leading state-of-the-art approaches have given us evidence of a substantial improvement (up to 4%) in accuracy, paving the way in this challenging area for actual clinical applications of machine learning-based approaches. Furthermore, a speeding-up of the training process was also achieved.

The main limitation of this work is the lack of testing in a clinical setting. Moreover, no study about models’ interpretability and explainability has been carried out, and we are aware this is a crucial aspect in medical-related systems. Prototyping and user evaluations are critical for attaining solutions that afford transparency, and the important results achieved must be considered as a starting point for setting up a system that could be employed in the clinical practice to allow users a quick and objective diagnosis. Future works will deal with testing in clinical settings and with a further improvement in classification performance by using different transformer architectures and ensemble strategies (e.g., based on stacking, boosting, and bagging). The explainability and interpretability of results will be also addressed by exploiting up-to-date methods [47]. Furthermore, the exploitation of transformer pre-processing on methods relying on multiple branches for global–local learning and on larger patches will be carried out.

## Figures and Tables

**Figure 1 sensors-23-04556-f001:**
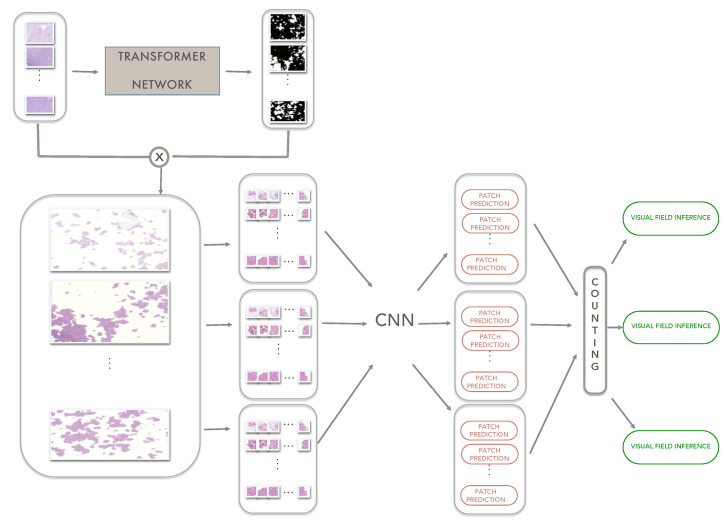
A schematic representation of the proposed pipeline exploiting a transformer architecture.

**Figure 2 sensors-23-04556-f002:**
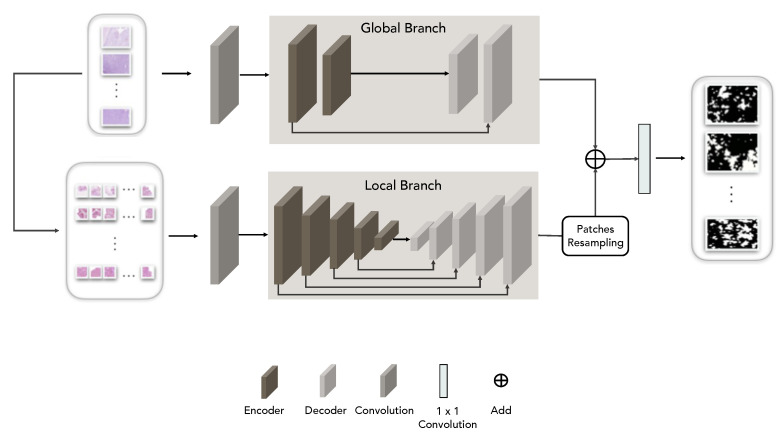
Schematic representation of the transformer architecture.

**Figure 3 sensors-23-04556-f003:**
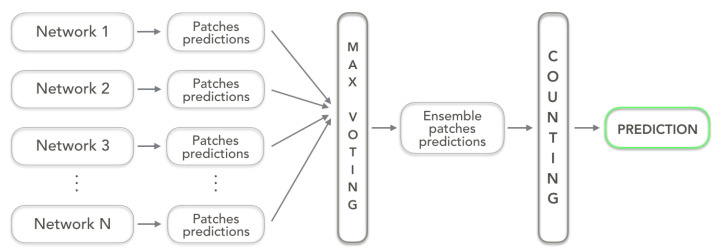
Max-voting ensemble architecture working on N networks.

**Figure 4 sensors-23-04556-f004:**
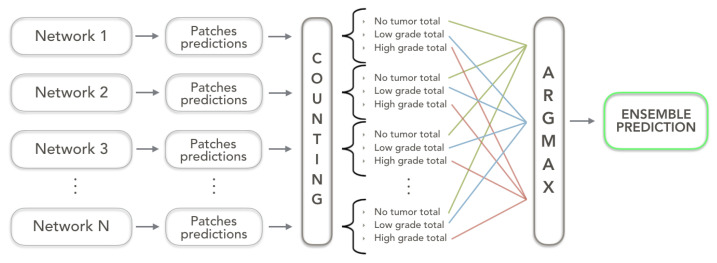
Argmax ensemble architecture working on N networks.

**Figure 5 sensors-23-04556-f005:**
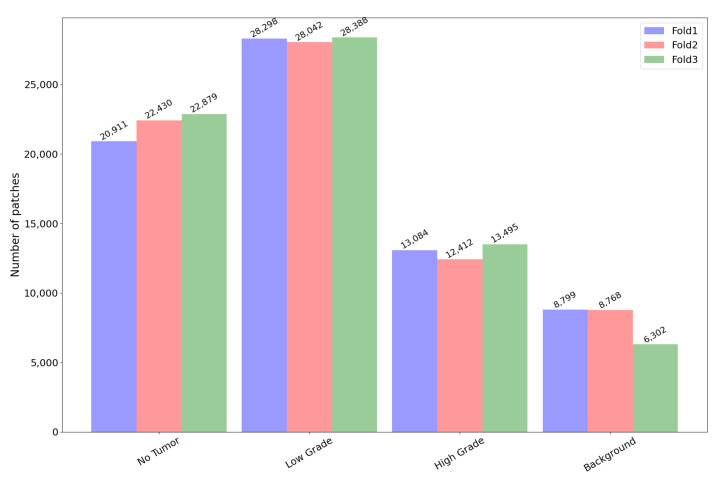
Patch distribution per fold and class.

**Figure 6 sensors-23-04556-f006:**
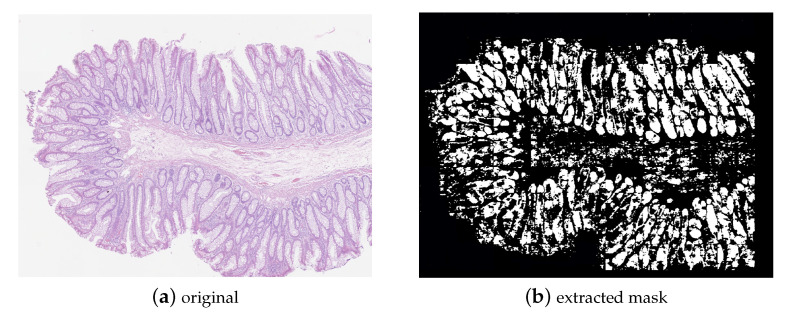
Glandular regions pointed out by the transformer network.

**Figure 7 sensors-23-04556-f007:**
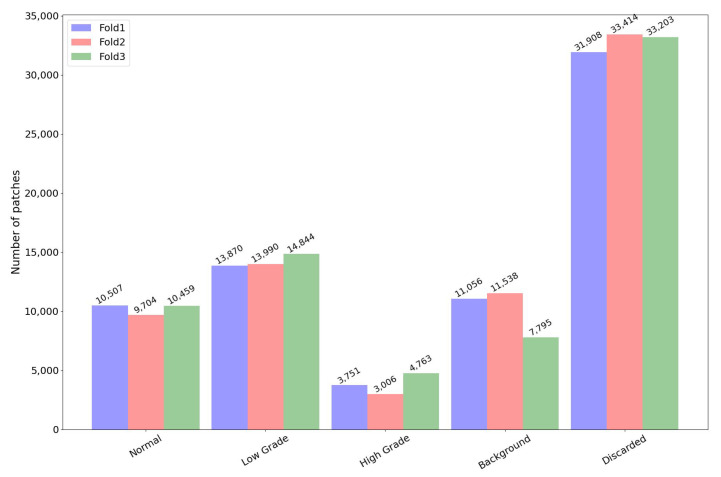
Patch distribution per fold and class when the transformer network was used.

**Table 1 sensors-23-04556-t001:** Distribution of visual fiels of different classes in CRC and Extended-CRC datasets.

Dataset	Normal	Low Grade	High Grade	Total
CRC	71	33	35	139
Extended-CRC	120	120	60	300

**Table 2 sensors-23-04556-t002:** Results on the Extended-CRC dataset while integrating transformer networks for pointing out discriminative regions and CNN architectures for tumor detection and grading.

Model	Average (%)	Weighted (%)	Average (%)	Weighted (%)
	(Binary)	(Binary)	(3-Classes)	(3-Classes)
T+DenseNet121	98.33 ± 1.25	98.17 ± 1.30	88.91 ± 3.81	86.37 ± 4.12
**T+EfficientNet-B1**	**99.67 ± 0.47**	**99.72 ± 0.39**	89.58 ± 4.17	87.50 ± 3.54
**T+EfficientNet-B2**	98.66 ± 0.95	98.74 ± 0.91	**89.92 ± 2.50**	**87.22 ± 2.08**
T+EfficientNet-B3	98.66 ± 0.95	98.89 ± 0.79	88.25 ± 2.55	84.44 ± 1.42
T+EfficientNet-B4	97.65 ± 0.95	98.06 ± 0.79	88.92 ± 3.62	84.72 ± 3.36
T+EfficientNet-B5	98.32 ± 1.72	98.46 ± 1.37	89.91 ± 5.42	88.05 ± 5.67
T+EfficientNet-B7	97.66 ± 1.24	97.91 ± 0.89	88.59 ± 1.30	84.44 ± 0.38
T+RegNetY6.4GF	94.30 ± 2.07	94.57 ± 1.75	85.89 ± 3.64	81.94 ± 5.11
T+RegNetY16GF	97.66 ± 1.88	97.62 ± 1.56	88.57 ± 3.93	85.54 ± 3.76
T+ResNet152	94.64 ± 2.04	94.82 ± 1.38	85.57 ± 1.31	81.36 ± 0.76
T+SE-ResNet50	95.31 ± 1.24	95.52 ± 0.83	85.22 ± 3.37	79.70 ± 6.34

**Table 3 sensors-23-04556-t003:** Average training time with and without integrating transformer networks for pointing out discriminative regions and CNN architectures for detection and grading.

Model	Average Training Time (min)	
	Transformer	No Transformer
DenseNet121	121.33	746.67
EfficientNet-B1	121.33	452.67
EfficientNet-B2	133.33	477.67
EfficientNet-B3	168.00	480.67
EfficientNet-B4	216.00	518.00
EfficientNet-B5	309.00	677.67
EfficientNet-B7	352.33	1188.00
RegNetY6.4GF	191.67	337.00
RegNetY16GF	349.00	699.33
Resnet152	199.00	493.67
SE-ResNet50	109.33	449.67

**Table 4 sensors-23-04556-t004:** Ensemble strategies and involved CNN architectures.

Label	Models	Strategy
E1	DenseNet121 EfficientNet-B7 RegNetY16GF	Max-Voting
E2	DenseNet121 EfficientNet-B7 RegNetY16GF SE-ResNet50	Max-Voting
E3	DenseNet121 EfficientNet-B7 RegNetY16GF RegNetY6.4GF	Max-Voting
E4	DenseNet121 EfficientNet-B7 RegNetY6.4GF	Max-Voting
E5	DenseNet121 EfficientNet-B2 RegnetY16GF	Max-Voting
E6	DeneNet121 EfficientNet-B2 RegNetY16GF	Max-Voting
E7	DeneNet121 EfficientNet-B2	Argmax
E8	DenseNet121 EfficientNet-B7 RegNetY16GF SE-ResNet50	Argmax
E9	EfficientNet-B7 RegNetY16GF SE-ResNet50	Argmax
E10	DenseNet121 EfficientNet-B2 RegNetY16GF	Argmax
E11	EfficientNet-B1 EfficientNet-B2 RegNetY16GF	Argmax
E12	EfficientNet-B1 EfficientNet-B2	Argmax

**Table 5 sensors-23-04556-t005:** Results in detection and grading by using ensembles of deep learning architectures. The leftmost column reports the ensembles by the labels introduced in Table 4. T+ point out the use of transformer networks for pre-processing.

Ensemble	Average (%)	Weighted (%)	Average (%)	Weighted (%)
	(Binary)	(Binary)	(3-Classes)	(3-Classes)
E1	95.65 ± 1.87	95.52 ± 1.85	86.90 ± 4.16	84.15 ± 3.81
E2	95.31 ± 2.48	95.68 ± 2.41	87.24 ± 3.37	83.88 ± 3.08
E3	95.31 ± 1.68	95.40 ± 1.89	87.23 ± 4.18	84.15 ± 4.10
E4	94.97 ± 1.62	95.12 ± 1.88	87.23 ± 1.18	84.15 ± 4.10
E5	95.98 ± 2.45	95.81 ± 2.72	86.90 ± 4.39	84.15 ± 3.81
E6	95.31 ± 2.34	95.40 ± 2.37	86.23 ± 3.37	83.32 ± 2.74
E7	95.65 ± 2.05	95.82 ± 2.23	87.91 ± 3.33	84.72 ± 3.43
E8	95.98 ± 2.15	95.95 ± 2.26	87.57 ± 3.75	84.71 ± 3.44
E9	97.32 ± 1.26	97.33 ± 1.57	88.24 ± 4.26	85.53 ± 3.76
T + E5	99.00 ± 0.82	99.02 ± 0.71	89.24 ± 4.09	87.49 ± 3.61
T + E7	99.33 ± 0.94	99.44 ± 0.79	89.58 ± 3.83	87.22 ± 3.87
T + E10	98.33 ± 1.25	98.46 ± 1.10	88.24 ± 4.10	85.52 ± 3.88
**T + E11**	**99.33 ± 0.94**	**99.44 ± 0.79**	**90.25 ± 3.74**	**88.06 ± 3.14**
T + E12	99.00 ± 0.82	99.02 ± 0.71	89.92 ± 3.00	87.49 ± 2.36

**Table 6 sensors-23-04556-t006:** Comparisons of results obtained with strategies in this paper and those in the literature.

Model	Average (%)	Weighted (%)	Average (%)	Weighted (%)
	(Binary)	(Binary)	(3-Classes)	(3-Classes)
Proposed				
**T+EfficientNet-B1**	**99.67 ± 0.47**	**99.72 ± 0.39**	89.58 ± 4.17	87.50 ± 3.54
**T+EfficientNet-B2**	98.66 ± 0.95	98.74 ± 0.91	89.92 ± 2.50	87.22 ± 2.08
**T + E11**	99.33 ± 0.94	99.44 ± 0.79	**90.25 ± 3.74**	**88.06 ± 3.14**
Previous works				
ResNet50 [25]	95.67±2.05	95.69±1.53	86.33±0.94	80.56±1.04
LR + LA-CNN [25]	97.67±0.94	97.64±0.79	86.67±1.70	84.17±84.17
CNN-LSTM [23]	95.33±2.87	94.17±0.79	82.33±2.62	83.89±2.08
CNN-SVM [20]	96.00±0.82	96.39±1.37	82.00±1.63	76.67±2.97
CNN-LR [20]	96.33±1.70	96.39±1.37	86.67±1.25	82.50±0.68
EfficientNet-B2 [26]	96.99 ± 2.94	96.65 ± 3.11	87.58 ± 3.36	85.54 ± 2.21
RegNetY-4.0GF [26]	95.64 ± 0.94	95.37 ± 1.52	84.55 ± 2.57	81.36 ± 1.43
RegNetY-6.4GF [26]	94.31 ± 2.48	94.26 ± 2.15	86.57 ± 2.12	83.58 ± 2.21

## Data Availability

Not applicable.

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
