# Peer review of "An Investigation about Modern Deep Learning Strategies for Colon Carcinoma Grading"

_sensors, 2023, doi:10.3390/s23094556_

Round 1
Reviewer 1 Report
The first idea is quit interesting. It is to use a transformer, which, thanks to the attention mechanism, can "correlate" not only nearby pixels, but also those located far away, to create binary masks. However, the implementation of the transformer model itself is described only in words, which is very, very difficult to perceive, I would like to see a diagram. In the dataset under study, there are more than 2 times fewer "high grade" cases than "low grade" cases. It seems that the imbalance of classes in a similar three-class segmentation problem is not very good. Please, give the comments. Table 3 shows the running time of models with preprocessing of the input image using a transformer, but I did not find in the article the time of similar models without a transformer. So there is nothing to compare with, but it is necessary. It wpuld be good not only to see the time, but to confirm in words that by cutting off unnecessary parts of the image, the work of the model is accelerated and how critical it is in this task. Table 6. The proposed methods are claimed to outperform SOTA methods. And even a comparison of the metrics of the proposed methods and SOTA is indicated. Nevertheless, it can be seen that the proposed methods, namely image preprocessing by a transformer, could be applied before the SOTA model. However, another one was chosen. That is, a comparison with the SOTA model ResNet50. And the proposed method is applied to EfficientNet and a couple more networks. Moreover, the same EfficientNet without a transformer gives an accuracy below ResNet50. Therefore, the question arises why it was considered and why was the transformer not applied to the best model of those with which the comparison is being made? Perhaps it only improves on some models, and makes it worse on others? This comparison seems a bit misleading to me. The author does not describe a strategy for dealing with inhomogeneities in the training sample. It occurs both at the high grade and low grade levels, as well as at the patch level. Authors should describe their strategy for dealing with learning heterogeneity or explain why such a strategy is not needed.
Reviewer 2 Report
There is a growing need for computer-aided approaches to cancer diagnosis and grading because they could eliminate inter- and intra-observer variability, speed up screening, favour early diagnosis, and then enhance the precision and consistency of the treatment planning processes. To improve colon cancer grading using histological pictures, this research systematically analyses the usage of convolutional neural networks and transformer topologies. Before final publication, this work must address some revisions/concerns.
1. The scientific value added/contributions of your paper are mentioned in your abstract as “To the best of our knowledge, this is the first attempt at using medical transformers and ensembling strategies for exploiting deep learning paradigms for automatic colon cancer diagnosis.” See following works, seems to do same type of job.
Paladini, E., Vantaggiato, E., Bougourzi, F., Distante, C., Hadid, A., & Taleb-Ahmed, A. (2021). Two ensemble-CNN approaches for colorectal cancer tissue type classification. Journal of Imaging, 7(3), 51.
Vantaggiato, E., Bougourzi, F., Distante, C., Hadid, A., & Taleb-Ahmed, A. (2021). Two Ensemble-CNNs Approaches for Colorectal Cancer Tissue Types Classification. J. Imaging, 7, 51.
2. Why Max-Voting/Argmax ensemble strategies are preferred for ensembling? Discuss. Have you compared other strategies?
3. What is the value of N (networks) in Figure 2. Max-Voting Ensemble architecture and Figure 3. Argmax Ensemble architecture.
4. Provide detailed architecture of CNN utilized.
5. 3-fold cross-validation was carried out in experiments? Why? Any argument.
6. Provide the details of hyperparameters and tuning.
7. Discuss the results achieved in abstract.
8. 6. Is there any limitation of framework? If yes, discuss.
9. Distribution of visual field of different classes in CRC and Extended CRC datasets seems imbalanced. Have authors utilized any data balancing methods such as SMOTE?
Minor editing of English language required.
